# Limited geographic genetic structure detected in a widespread Palearctic corvid, *Nucifraga caryocatactes*

Kimberly M. Dohms and Theresa M. Burg

Department of Biological Sciences, University of Lethbridge, Lethbridge, AB, Canada

## ABSTRACT

The Eurasian or spotted nutcracker (*Nucifraga caryocatactes*) is a widespread resident corvid found throughout the Palearctic from Central Europe to Japan. Characterized by periodic bouts of irruptive dispersal in search of *Pinus* seed crops, this species has potential for high levels of gene flow across its range. Previous analysis of 11 individuals did not find significant range-wide population genetic structure. We investigated population structure using 924 base pairs of mitochondrial DNA control region sequence data from 62 individuals from 12 populations distributed throughout the nutcracker's range. We complemented this analysis by incorporating additional genetic data from previously published sequences. High levels of genetic diversity and limited population genetic structure were detected suggesting that potential barriers to dispersal do not restrict gene flow in nutcrackers.

## INTRODUCTION

In Eurasia, phylogeographic studies of many widespread vertebrate species have revealed a variety of geographical patterns of population structure influenced by current and historical landscapes, with little overall consensus among species. Using mitochondrial DNA, east–west splits have been documented for a variety of vertebrates including bats (*Flanders et al., 2009*), and several avian species (e.g., Eurasian magpie (*Pica pica*; *Kryukov et al., 2004*), rook (*Corvus frugilegus*; *Haring, Gamauf & Kryukov, 2007*), and red-breasted flycatcher (*Ficedula parva*; *Zink et al., 2008*)). For other species, multiple splits have occurred (e.g., root vole (*Microtus oeconomus*; *Brunhoff et al., 2003*) and reed bunting (*Emberiza schoeniclus*; *Zink et al., 2008*)), or peninsula populations are isolated (e.g., great bustard (*Otis tarda*; *Pitra, Lieckfeldt & Alonso, 2000*)). In contrast, little population structure has been detected in some widespread species, such as otters (*Lutra lutra*; *Ferrando et al., 2004*) and several avifauna species (e.g., great spotted woodpecker (*Dendrocopos major*; *Zink, Drovetski & Rohwer, 2002*), common sandpiper (*Actitis hypoleucos*; *Zink et al., 2008*), and Eurasian magpie (*Pica pica*; *Zhang et al., 2012*)). Some of the observed phylogeographic patterns have been explained by post-glacial colonization from single or multiple refugia, but may also be influenced by barriers to dispersal, such as

Corresponding author
Kimberly M. Dohms,
kim.dohms@uleth.ca

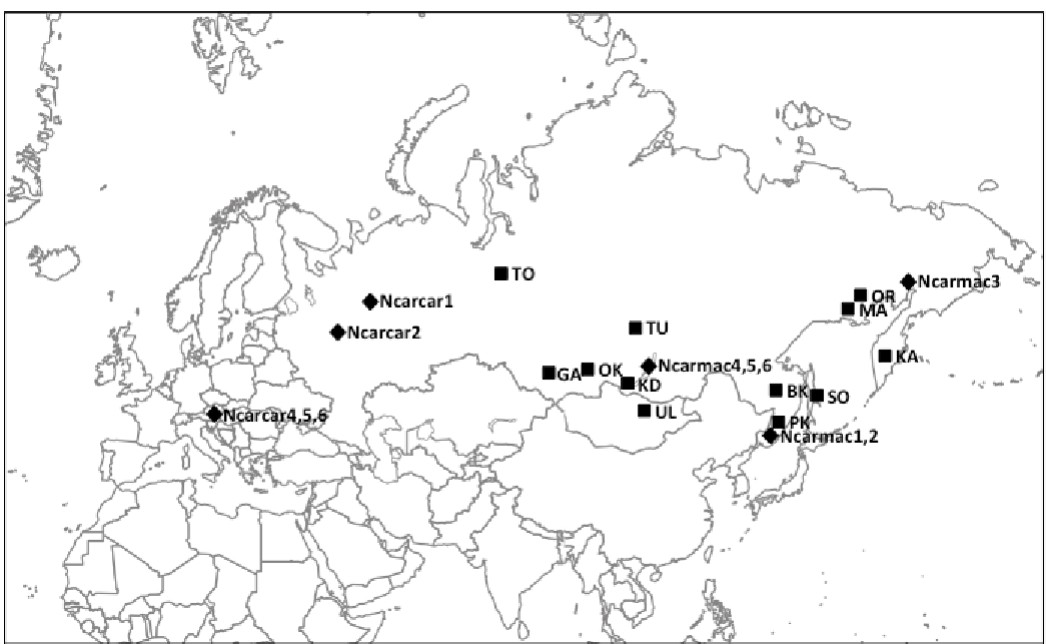

**Figure 1 Nutcracker tissue sample locations throughout Eurasia.** Black squares denote locations; corresponding abbreviations are labelled beside squares. Refer to Table 1 and Table S1 for further location information. Black diamonds denote locations of previously published partial control region sequences obtained from GenBank with corresponding sample codes from *Haring, Gamauf & Kryukov (2007)*.

mountain ranges (e.g., Ural Mountains), large areas of inhospitable habitat (e.g., Tibetan Plateau), or large bodies of water.

The Eurasian nutcracker (*Nucifraga caryocatactes*, Linnaeus, 1758) is a corvid with a widespread Palearctic distribution. Although generally classified as a resident species of continental coniferous forests, nutcrackers are known to irruptively disperse to take advantage of mast conifer seed crops (*Haring, Gamauf & Kryukov, 2007*), similar to its North American sister species, Clark's nutcracker (*N. columbiana*; *Tomback, 1998*). Strong geographic genetic structure has not been found in Clark's nutcracker, despite numerous potential physical barriers to dispersal and thus gene flow (*Dohms & Burg, 2013*). A previous study by Haring and colleagues (*2007*) found no population structure in *N. caryocatactes* throughout Eurasia. However, *Haring, Gamauf & Kryukov (2007)* only used 11 specimens, thus additional data may shed further light on nutcracker population genetic structure.

In this study, we use a highly variable and rapidly evolving mitochondrial DNA marker, the control region, to further investigate population structure of *N. caryocatactes* in Eurasia. Based on the ecology of this species, we predict little range-wide population genetic structure.

## MATERIALS & METHODS

### Samples

Tissue samples ($n = 62$) collected throughout the Eurasian nutcracker's range (Fig. 1) were acquired from the Burke Museum of Natural History and Culture at the University

of Washington (Table S1). Previously published control region (CR) sequences ($n = 11$) were obtained from GenBank (EU070770 and EU070886–EU070895; *Haring, Gamauf & Kryukov, 2007*).

### DNA extraction, PCR amplification, and sequencing

DNA from muscle samples stored in ethanol or lysis buffer was extracted using a modified Chelex extraction protocol (*Walsh, Metzger & Higuchi, 1991*; *Burg & Croxall, 2001*). A 924 bp fragment starting at position 46 of the control region (CR; *Saunders & Edwards, 2000*) was amplified using two primers: L46 SJ (5′-TTT GGC TAT GTA TTT CTT TGC-3′; developed for Steller's jay (*Cyanocitta stelleri*; T Birt & K Lemmen, 2005, unpublished data)) and H1030 JCR 18 (5′-TAA ATG ATT TGG ACA ATC TAG G-3′; developed for *Aphelocoma* jays (*Saunders & Edwards, 2000*)). DNA was amplified in a Master gradient thermocycler (Eppendorf) in 25 µL reactions with 1x goTaq Flexi buffer (Promega), 2.5 mM MgCl$_2$, 200 µM dNTP, 0.4 µM of each primer, and 1 unit goTaq Flexi taq polymerase (Promega). DNA sequencing was performed on an ABI 3730xl DNA Analyzer at McGill University and Génome Québec Innovation Centre.

### Alignment and analysis

We edited and aligned sequences from chromatograms and an overlapping subset of 305 bp from previously published CR sequences from GenBank (*Haring, Gamauf & Kryukov, 2007*) using MEGA v5.0 (*Tamura et al., 2007*). Two unrooted statistical parsimony networks (95% probability) were constructed with TCS v1.21 (*Clement, Posada & Crandall, 2000*): one for the samples sequenced as part of this study (924 bp) and a second network for the 305 bp common fragment (this study; *Haring, Gamauf & Kryukov, 2007*). We calculated the number of haplotypes ($H_n$), haplotype diversity ($H_d$), and nucleotide diversity ($\pi$) for museum samples using DnaSP v5.10 (*Rozas et al., 2003*).

## RESULTS

### Genetic analyses

We sequenced and aligned the 924 bp control region (CR) sequences for 62 individuals from 12 populations (Table 1; GenBank accession nos. KJ999615–KJ999676). We aligned 11 additional GenBank sequences (*Haring, Gamauf & Kryukov, 2007*) with sequences from our samples and obtained a 305 bp area of overlap. Statistical parsimony networks did not suggest strong geographic structure for the 924 bp sequence (Fig. 2), nor for the larger dataset using the overlapping 305 bp fragment for all 73 individuals (Fig. 3). Ncarcar5 could not be connected to the network in the larger dataset, which was also found by *Haring, Gamauf & Kryukov (2007)*.

For the 62 individuals we sequenced, we found 45 unique haplotypes and high levels of genetic diversity in most populations (Table 1). We found 57 polymorphic sites within the 924 bp sequence and 22 within the 305 bp sequence. Haplotype diversity for the 924 bp sequence varied from 0.000 (Ola River headwaters (OR)) to 1.000 (five populations), and all but the OR population had a haplotype diversity equal to or greater than 0.800

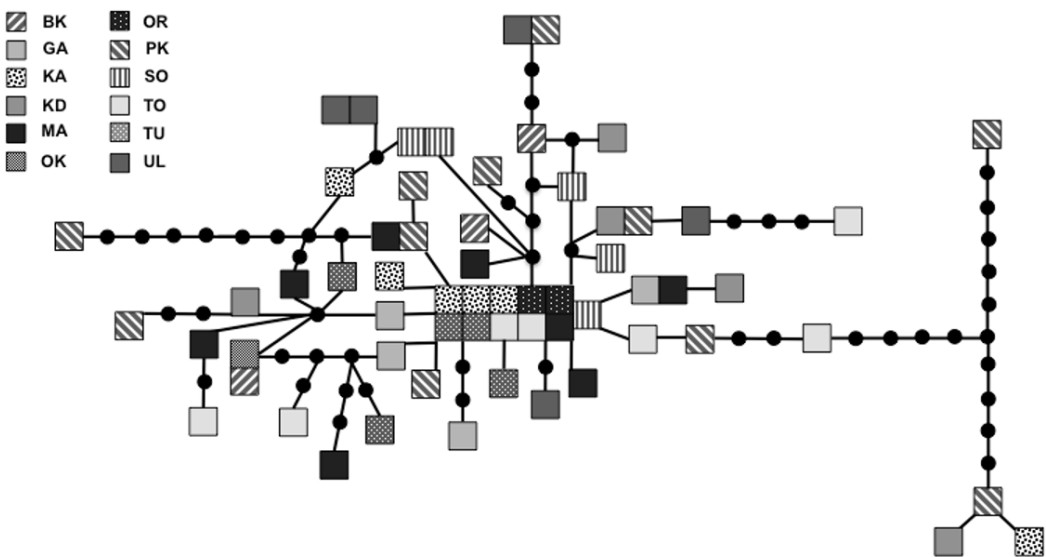

**Figure 2 Statistical parsimony network for 924 bp mitochondrial DNA sequence.** Statistical parsimony network of *Nucifraga caryocatactes* for 924 bp of the mitochondrial DNA control region sequenced from museum samples ($n = 62$). Each square represents one individual and colours correspond to author-defined populations, as per figure legend. Circles indicate inferred haplotypes. Refer to Table 1 for population abbreviations.

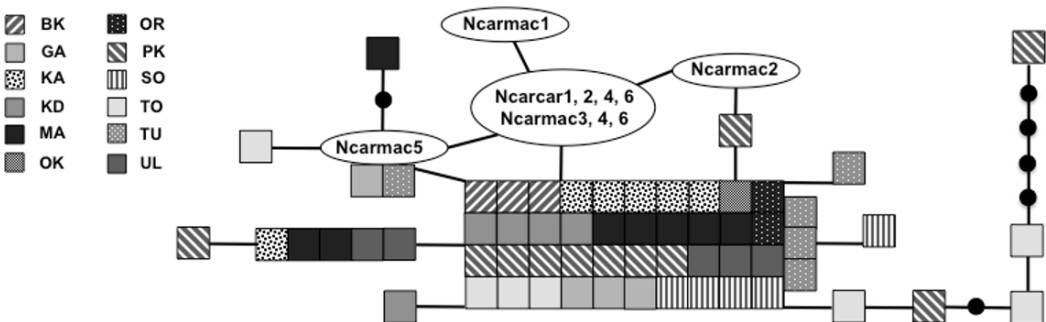

**Figure 3 Statistical parsimony network of 305 bp mitochondrial DNA sequence.** Statistical parsimony network of *Nucifraga caryocatactes* for overlapping sequences of 305 bp of the mitochondrial control region (Domain II) sequenced from museum samples ($n = 62$) and GenBank sequences ($n = 11$; *Haring, Gamauf & Kryukov, 2007*). Each coloured square represents one individual and colours correspond to author-defined populations. Black solid circles indicate inferred haplotypes. Open circles represent haplotypes; text in circles represents GenBank sequences as per Fig. 1 and *Haring, Gamauf & Kryukov (2007)*. Refer to Table 1 for population abbreviations found in the legend.

(Table 1). Nucleotide diversity ranged from 0.00000 (OR) to 0.00993 (PK; Table 1). Overall haplotype diversity ($H_d$) 0.967 and nucleotide diversity ($\pi$) was 0.00537.

## DISCUSSION

As predicted, analyses of Eurasian nutcracker mitochondrial DNA control region sequences did not detect significant population genetic structure. All populations except nutcrackers from the Ola River headwaters (OR; $n = 2$) exhibited high haplotype diversity

Table 1  **Diversity of a 924 bp mitochondrial DNA sequence.** Haplotype diversity for a 924 bp fragment of mtDNA control region from 12 populations of *Nucifraga caryocatactes* throughout Eurasia.

| Population | Location | $n$ | $H_n$ | $H_d$ | $\pi$ |
|---|---|---|---|---|---|
| BK | Badzhal'skiy Krebet, Russia | 3 | 3 | 1.000 | 0.00361 |
| GA | Gorno-Altaysk, Russia | 4 | 4 | 1.000 | 0.00328 |
| KA | Kamchatka, Russia | 6 | 4 | 0.800 | 0.00369 |
| KD | Irkutsk Oblast, Russia | 5 | 5 | 1.000 | 0.00523 |
| MA | Magadanskaya Oblast, Russia | 8 | 8 | 1.000 | 0.00397 |
| OK | Kyzyl, Russia | 1 | 1 | - | - |
| OR | Ola River headwaters, Russia | 2 | 1 | 0.000 | 0.00000 |
| PK | Primorsky Kray, Russia | 11 | 11 | 1.000 | 0.00993 |
| SO | Sakhalinksya Oblast, Russia | 5 | 4 | 0.900 | 0.00195 |
| TO | Tyumenskaya Oblast, Russia | 7 | 6 | 0.952 | 0.00622 |
| TU | Irkutsk Oblast, Russia | 5 | 4 | 0.900 | 0.00326 |
| UL | Ulaanbaatar, Mongolia | 5 | 4 | 0.900 | 0.00611 |
| **Overall** | | **62** | **45** | **0.967** | **0.00537** |

**Notes.**

$n$, number of individuals in population; $H_n$, number of haplotypes; $H_d$, haplotype diversity; $\pi$, nucleotide diversity within the population.

and relatively high nucleotide diversity. No geographic clustering was observed in statistical parsimony networks, even when integrating samples from the western part of the range. Despite potential barriers to dispersal for this species, such as isolation on an island (e.g., Sakhalin Oblast (SO)) or peninsula (e.g., Kamchatka (KA)), most populations of *N. caryocatactes* do not appear to be geographically differentiated from each other, likely due to gene flow during irruptive dispersal in search of mast pine seed crops. Overall, our work supports that done by *Haring, Gamauf & Kryukov (2007)* where no significant split was seen between the east and west for nutcrackers and is similar to the pattern found in *N. caryocatactes* sister species, *N. columbiana* (*Dohms & Burg, 2013*).

Compared to *Haring, Gamauf & Kryukov (2007)*, our study found a higher level of haplotype diversity ($H_d = 0.967$ vs 0.844 and $\pi = 0.00537$ vs 0.00279). This may be due to the portion of control region sequenced and the larger sample sizes used in this study. The sequences obtained from our samples are predominantly composed of domains I and II of the mtDNA control region (*Saunders & Edwards, 2000*), whereas *Haring, Gamauf & Kryukov (2007)* sequenced primarily domain II, which is considered less variable (*Ruokonen & Kvist, 2002*).

*Haring, Gamauf & Kryukov (2007)* state that low genetic diversity may suggest a bottleneck in this species and a single glacial refugium. With high levels of nucleotide and haplotype diversity, our findings do not support a historical bottleneck but rather point toward two possible scenarios: multiple refugia with gene flow or a single refugium with large population size during expansion. Expansion from multiple refugia with gene flow after colonization can produce similar genetic patterns to those in species that expanded slowly from a single refugium with a large population size, retaining high genetic diversity and limited geographic structuring of populations (*Hewitt, 2004*). Given the

dispersal potential of nutcrackers, it is possible that a large population expanded out of a single refugium, but it is equally plausible that expansion occurred out of multiple refugia with subsequent gene flow between geographically distinct populations due to irruptive dispersal events. The multiple refugia scenario could have produced the large number of haplotypes, often with high levels of sequence divergence pattern seen here. For example, individuals from Primorsky Kray (PK) are found scattered throughout the parsimony network (Fig. 2), in some cases with a large number of mutations between PK individuals and other haplotypes, yet found clustered with geographically distant individuals from Irkutsk Oblast (KD) and KA. This level of divergence is often associated with isolation in and subsequent colonization from multiple refugia (*Hewitt, 2004*). With the high haplotype diversity across all populations, it is not possible to determine which population(s), if any sampled here, may be in the location of the original refugium or refugia. Without additional present day samples from the Alps and Himalaya Mountain ranges, it is difficult to tell using genetic signatures if these areas served as refugia for nutcrackers during the LGM.

Our findings do not support a single refugium in the Altai Mountains of southern Mongolia, as postulated by *Haring, Gamauf & Kryukov (2007)*. Rather, our data show highly divergent haplotypes which could be the result of prolonged isolation in multiple refugia. Scots pine (*Pinus sylvestris*), an important source of food for nutcrackers, is thought to have survived in refugia near the Alps (*Naydenov et al., 2007*) and in the east, unglaciated portions of the Himalayas could have served as a refugium for high latitude species (*Zhuo, Baoyin & Petit-Maire, 1998*; *Owen, Finkel & Caffee, 2002*). Alternatively, high levels of plant endemism have been found in the mountains of southern and eastern China, suggestive of long-term suitable habitats (*Zhuo, Baoyin & Petit-Maire, 1998*). Nutcrackers may have survived in these bands of suitable habitat in the southwest and southeast areas of Eurasia and expanded northward from multiple refugia as glaciers retreated.

## CONCLUSIONS

Overall, Eurasian nutcrackers exhibit limited geographic genetic structure throughout their range, as might be expected from a resident bird with irruptive dispersal patterns. Our study found high genetic diversity, which suggests that a population bottleneck has not occurred in this species as previously hypothesized. A more detailed phylogeographical study could include additional genetic sampling from northern and southern parts of *N. caryocatactes*' range to further investigate structure across the range of this species.

## ACKNOWLEDGEMENTS

We gratefully acknowledge the assistance of Sharon Birks and the Burke Museum of Natural History and Culture—University of Washington; without tissue samples from their collection, this study would not have been possible.

### Funding
Funding for this project was provided by a Natural Science and Engineering Research Council (NSERC) Discovery Grant (TMB) and Post-Graduate Scholarship D (KMD) and Alberta Innovates (AI) New Faculty Award (TMB) and Graduate Scholarship (KMD). The funders had no role in study design, data collection and analysis, decision to publish, or preparation of the manuscript.

### Grant Disclosures
The following grant information was disclosed by the authors:
Natural Science and Engineering Research Council (NSERC): Discovery Grant (TMB) and Post-Graduate Scholarship D (KMD).
Alberta Innovates (AI): New Faculty Award (TMB) and Graduate Scholarship (KMD).

### Competing Interests
The authors declare there are no competing interests.

### Author Contributions
- Kimberly M. Dohms conceived and designed the experiments, performed the experiments, analyzed the data, wrote the paper, prepared figures and/or tables, reviewed drafts of the paper.
- Theresa M. Burg conceived and designed the experiments, contributed reagents/materials/analysis tools, wrote the paper, reviewed drafts of the paper.

### Animal Ethics
The following information was supplied relating to ethical approvals (i.e., approving body and any reference numbers):

We did not handle any live vertebrate animals during this study. All animals sequenced here were sequenced from tissue samples provided by the Burke Museum, thus animal care approval was not required for this specific study.

### DNA Deposition
The following information was supplied regarding the deposition of DNA sequences:

GenBank: NUCA.sqn BK001 KJ999615, NUCA.sqn BK002 KJ999616, NUCA.sqn BK003 KJ999617, NUCA.sqn GA001 KJ999618, NUCA.sqn GA002 KJ999619, NUCA.sqn GA003 KJ999620, NUCA.sqn GA004 KJ999621, NUCA.sqn KA001 KJ999622, NUCA.sqn KA002 KJ999623, NUCA.sqn KA003 KJ999624, NUCA.sqn KA004 KJ999625, NUCA.sqn KA005 KJ999626, NUCA.sqn KA006 KJ999627, NUCA.sqn KD001 KJ999628, NUCA.sqn KD002 KJ999629, NUCA.sqn KD003 KJ999630, NUCA.sqn KD004 KJ999631, NUCA.sqn KD005 KJ999632, NUCA.sqn MA001 KJ999633, NUCA.sqn MA002 KJ999634, NUCA.sqn MA003 KJ999635, NUCA.sqn MA004 KJ999636, NUCA.sqn MA005 KJ999637, NUCA.sqn MA006 KJ999638, NUCA.sqn MA007 KJ999639, NUCA.sqn MA008 KJ999640, NUCA.sqn OR001 KJ999641, NUCA.sqn OR002 KJ999642, NUCA.sqn PK001 KJ999643,

NUCA.sqn PK002 KJ999644, NUCA.sqn PK003 KJ999645, NUCA.sqn PK004 KJ999646, NUCA.sqn PK005 KJ999647, NUCA.sqn PK006 KJ999648, NUCA.sqn PK007 KJ999649, NUCA.sqn PK008 KJ999650, NUCA.sqn PK009 KJ999651, NUCA.sqn PK010 KJ999652, NUCA.sqn PK011 KJ999653, NUCA.sqn SO001 KJ999654, NUCA.sqn SO002 KJ999655, NUCA.sqn SO003 KJ999656, NUCA.sqn SO004 KJ999657, NUCA.sqn SO005 KJ999658, NUCA.sqn TO001 KJ999659, NUCA.sqn TO002 KJ999660, NUCA.sqn TO003 KJ999661, NUCA.sqn TO004 KJ999662, NUCA.sqn TO005 KJ999663, NUCA.sqn TO006 KJ999664, NUCA.sqn TO007 KJ999665, NUCA.sqn TU001 KJ999666, NUCA.sqn TU002 KJ999667, NUCA.sqn TU003 KJ999668, NUCA.sqn TU004 KJ999669, NUCA.sqn TU005 KJ999670, NUCA.sqn UL001 KJ999671, NUCA.sqn UL002 KJ999672, NUCA.sqn UL003 KJ999673, NUCA.sqn UL004 KJ999674, NUCA.sqn UL005 KJ999675, NUCA.sqn OK001 KJ999676.

## Supplemental Information

Supplemental information for this article can be found online at http://dx.doi.org/10.7717/peerj.371.

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
