# Peer review of "Limited geographic genetic structure detected in a widespread Palearctic corvid, Nucifraga caryocatactes"

_PeerJ, doi:10.7717/peerj.371_

## Round 0.1 · original submission · Major Revisions

Dear authors
Thank you for submitting your manuscript to our journal. As you see our reviewers suggest a thorough revision of your ms. If you are willing to do so, we would be happy to reconsider your revised manuscript.

Michael Wink

Reviewer 1 ·

Basic reporting

The manuscript adheres to all PeerJ policies and follows the Instructions for Authors.

Experimental design

The paper reports new findings within the Aims & Scope of the Journal. All methods are clearly documented. Sampling for species-distribution modeling is inadequate, especially for the Siberian part of the range.

Validity of the findings

The data basis has been broadened compared to a previous article on the phylogeography of Nucifraga caryocatactes. The reconstructed current distribution is far from realistic, thus any extrapolation to another era is not recommend. Necessarily, conclusions from such data are without justification.

Additional comments

Your provide a well-written phylogeographic study which widens our knowledge a lot. If you lack the data to model the distribution adequately, better refrain from incorporating SDM's.

·

Basic reporting

The language is clear and flawless, the manuscript is well written.
Introduction and background are well outlined to set the study aims and results in a broader context including citations of relevant literature.
Figs 1-4 are relevant and self-explanatory; Figure 5 is debatable because the use of SDM analysis requires a convincing justification (see comments to authors); Table 2 is not relevant because of its low (doubtful) informative value (see comments to authors).

Experimental design

The submission fits the journal scope and addresses an interesting and relevant research question. Aims and expectations are well formulated.
Genetic methods are well outlined the analyses are up to current standard.
The sampling has limitations but is generally good in the central and eastern part of the range (see comments to authors). All sequence data are based on vouchered samples which is a clear merit of the study.
SDM analysis is not well outlined because basic information on the presence record data are missing. Of course, it cannot be expected to provide a full table of all 503 records. But more information of the distribution of presence records across the current range of the study species are essential to judge whether the strong deviations of the modeled distribution from the real distribution range is a sampling artifact. Generally, if the authors cannot provide a real strong justification for their SDM approach that would outweigh the strong deviations the SDM analysis should be reconsidered.

Validity of the findings

The genetic data are robust and statistically sound. These should be the focus of the study - limitations of sampling and sequence length of the combined analyses of own and GenBank sequences should be discussed.
Own sequences must be deposited at GenBank - no such statement is given so far in the text nor in the supplemental Table S1.
The SDM analysis is questionable because of the strong deviations of modeled extant distribution from the real distribution range (see detailed comment to authors). If these results should be seriously presented and discussed then a detailed assessment of possible sources of error (distribution of presence records, model settings etc.) and a much more careful interpretation of the results (two refuges) must be provided by the authors.To my opinion taking suboptimal modelling results for granted (as a base for LGM modelling and in the discussion) is not a proper application of this method.

Additional comments

I have reviewed an earlier version of this paper and in fact I have the same objections to some basic limitations (if not flaws) of the study that have not been addressed in the present version yet.

Limitation 1: The own sample set is not “distributed throughout the nutcracker´s range” (abstract) but it comprises only samples from eastern and central Palearctic populations. This is in fact a strong limitation of the explanatory power of the data set with respect to two possible refuge areas because all analyses based solely on the own data do not comprise samples from the western Palearctic (the region of the hypothesized western refuge in the Alps).
The extended data set including western samples from the Haring et al. (2007) study however is limited to 305 bp only. The inclusion of these sequence data was reasonable, but necessarily implied cutting down the sequence alignment and thus again reduces the informativeness of genetic variation when potential east-west splits shall be analysed. This limitation must be addressed in the discussion.
Actually, the conclusions drawn from the genetic results as outlined in l. 149-175 seem plausible even considering the limitations of the data set. The sampling though restricted to the central and eastern part of the range is reasonable, particularly considering that capture of this species might not be that easy and genetic samples are not easily available in great numbers. Furthermore all genetic samples are vouchered which is another merit of the study. So the focus of the paper should be clearly on the genetic analysis.

Limitation 2: Distribution modeling has recently become sort of a “must have” in any kind of phylogeographic study whether it makes sense or not. I still doubt whether the modeling results of this study make sense at all: The modeled extant distribution strongly deviates from the true distribution of Nucifraga caryocatactes, only the “western and southeastern portions of the range” were predicted as suitable habitat (as rightly stated in l. 133), but the vast northern distribution range was not predicted as suitable by the model. I do not see, that such strong deviation from the true extant distribution justifies the conclusion that the “current model was highly suitable for backcasting to the paleodistribution model”. No information is given on the actual database of 503 presence records (which I would presume is a good base, so why does the model come to such strongly deviating predictions?) and whether the modeled distribution was due to a sampling artifact. Moreover, if this is not the case and presence records from the northern range were included (and not strongly underrepresented) in the model – then why these strong deviations? Actually, the complete lack of any critical evaluation of the modeling approach and the crucial results is disappointing.
Finally, there is no evidence from the genetic data for an east-west population subdivision in the nutcracker neither from the Haring et al. (2007) study nor from the current analysis. Despite of this lack of evidence the scenario of separation in western (Alps) and eastern (Himalayan) montane refuge areas is widely discussed in l. 176-189. The whole approach does not seem convincing to me and without any critical assessment of these results I do not see a real gain of information from the SDM analysis – and honestly without any convincing justification for the SDM analysis despite these deviating results I suggest that they should be disclaimed.

Limitation 3: Genetic diversity parameters for populations of n< 5 (Table 2) are not informative and I consider this analysis a true flaw of the study (populations BK, GA). Local samplings are really rather limited throughout the sampled range, so I strongly recommend focussing the presentation of results to rather descriptive parameters in Table 1 and discarding Table 2.

·

Basic reporting

The article is concise and addresses its purpose using relevant information and media. I have just some minor comments in this area:

- Please check the best format for scientific names according to the instructions - "Authors are encouraged to provide taxonomic authors of Linnean binomials when first used in the text, particularly for taxa that are the focus of the paper in question"

- Figure 1 could be integrated in figure 5 (present distribution). It would decrease the number of figures and also allow us to see easier how representative is the sample (especially because the projections are different between figures).

Experimental design

Figure 2 & 3 - I find the networks confusing due to:
1. The use of many colors. As a representative of the fraction of male caucasians that are color blind, I must say this is not the best option. I would group locations in major regional classes, probably with a gradient of colors or levels of gray correlated with distance to main refugia.
- It is difficult to see the frequencies of each location using this "squares" approach. Although piecharts also have problems, they seem better for this kind of data

I advise the following solution:
- Simple pie chart network with pies corresponding to each class and size correlated with haplotype frequency.
- The classes should be smaller. There is no point in identifying each location. Instead, I find it more crucial to have regional classes.

Validity of the findings

The authors prove, using niche modelling, that Altai mountains (S Mongolia) may not be the only suitable refugia in Eurasia, with areas in Europe (Alps) and Himalaya with high suitability and extensive suitable areas in Europe, Himalaya, China and Japan. This a valid and important finding of the article, although it is considered as a secondary finding in the article, used more as support information to discuss the genetic results.

But for the first objectives (current genetic structure and past demographic patterns) the experimental design has some flaws. The accurate sampling of genetic diversity all over the distribution is necessary to identify possible hotspots of genetic diversity that may be related to expansion patterns from refugia. Their own findings concerning the past higher suitability of southern areas of the distribution gives even more support for the importance of sampling these areas, where haplotype diversity is supposed to be higher.

But, unfortunately, when we overlap the sample location to the current distribution, most of these need to be considered as "recent" locations, due to expansion, according to the author's rationale. The sampling locations that can be considered "old" and representative of refugia are few. Therefore, many of the inferences and test of hypothesis regarding the number of refugia and the occurrence of gene flow can't be conclusive.

The authors show high haplotype diversity and low genetic structure. The predictions of the multiple refugia hypothesis are the occurrence of haplotype structure, especially related to distance, possibly with distinct clades. If current gene flow is high, you would observe these haplotypes disseminated throughout the sample locations. The single refugia hypothesis would predict that most haplotypes were related to each other (e.g. star-like pattern) and if gene flow is high you would still see all haplotypes disseminated and not a decrease of genetic diversity with distance to refugia hotspots.

The control region, due to its higher variability, may not be the best marker for testing these hypothesis and others less variable markers (e.g. ND2) would probably add more insights to interpret these patterns.

Therefore, I advise the authors to be even more careful with the interpretation of their findings, since I do not think they can be amalgamated into just 2 hypothesis with high/low gene flow and single/multiple refugia in the same hypotheses.

---

## Round 0.2 · accepted · Accept

Dear author
thanks for the revision. The paper is now ready for publishing

Reviewer 1 ·

Basic reporting

No Comments.

Experimental design

No Comments.

Validity of the findings

No Comments.

Additional comments

You successfully revised the manuscript and followed the reviewers' recommendations as far as they did not exclude each other. Naming of the species (e.g. "Nucifraga caryocatactes, Linnaeus, 1758") still does not follow standard and should be adjusted in the final version of the manuscript. See http://www.worldbirdnames.org/n-vireos.html for the correct use of comma and parentheses as well as current taxonomy. If you follow the latter, N. caryocatactes is not the sister species of N. columbiana. I would recommend to replace the term "sister species" in ll. 29 and 91 by "congener" or "counterpart" to escape from a larger debate whether or not to recognize N. multipunctata as a full species and thus the sister species of N. caryocatactes.

---

## Author Rebuttal · Round 0.2

**Department of Biological Sciences**
4401 University Drive W
Lethbridge, Alberta, Canada
T1K 3M4

Phone   1-403-332-5213
Fax      1-403-329-2082
Email   kim.dohms@uleth.ca

27 March 2014

Dear Dr. Michael Wink,

Thank you and the three reviewers for constructive and helpful feedback on our manuscript entitled, "Limited geographic genetic structure detected in a widespread Palearctic corvid, *Nucifraga caryocatactes*" (Reference #2013:10:945:0:0:REVIEW) Please find below our response to the comments provided.

### Species Distribution Modeling (SDM)
Reviewers 1 & 2 recommend removal of this from the manuscript and we agree with this feedback. We have removed all sections regarding this analysis from the Materials & Methods and Results, Figure 5 (which documents the results of the SDM), and references to it in the Introduction and Discussion.

### Reviewer-specific comments
### Reviewer 1
Comments regarding SDM are addressed in general changes above.

### Reviewer 2 (Martin Paeckert)
"*Figure 5 is debatable because the use of SDM analysis requires a convincing justification*" *(and subsequent comments).*
As per the above changes, we have removed the SDM analysis and thus, Figure 5.

"*Table 2 is not relevant because of its low (doubtful) informative value.*" *(and subsequent comments).*
We agree with Dr. Paeckert's comment in light of a paper published by R.M. Harding (1997; Institute for Mathematics and Its Applications, Vol. 87, p.15) that suggests $F_{ST}$ values comparing populations with n<10 individuals (of limited polymorphism) are of low informative value. We have removed this table as well as the PCA analysis (based on $F_{ST}$ values) and associated methods, results, and discussion related to these measures. We have chosen to focus on other genetic analyses such as haplotype diversity and the statistical parsimony network.

"*Own sequences must be deposited at GenBank - no such statement is given so far in the text nor in the supplemental Table S1.*"
We will deposit sequences in GenBank upon publication of the manuscript. We have added a line in the results (Lines 49-50) and a column has been added to supplemental Table S1 to reflect this.

"*The inclusion of these sequence data was reasonable, but necessarily implied cutting down the sequence alignment and thus again reduces the informativeness of genetic variation when*

*potential east-west splits shall be analysed. This limitation must be addressed in the discussion.*"
We have added a note about the number of variable sites within this sequence in the results (Line 85) and mentioned its limitation in the discussion.

"*…western (Alps) and eastern (Himalayan) montane refuge areas is widely discussed in l. 176-189.*"
This section has been removed in response to this comment and Dr. Lopes' comments below.

**Reviewer 3** (Ricardo Lopes)
*"Please check the best format for scientific names according to the instructions"*
We have added "Linnaeus, 1758" to the first reference of *Nucifraga caryocatactes* in the Introduction on Line 31.

*"Figure 1 could be integrated into Figure 5…"*
Figure 5 has been removed from the manuscript entirely (see above), thus this revision was not made.

*"Figure 2 & 3 - …"*
We have revised the colour in these networks to be grayscale (please see figure legend for explanation) for ease of viewing. However, as the other reviewers did not take issue with the block and line design of the network and we feel it is informative, we have left the design as is and not changed it to a pie chart format.

"*The sampling locations that can be considered "old" and representative of refugia are few. Therefore, many of the inferences and test of hypothesis regarding the number of refugia and the occurrence of gene flow can't be conclusive.*"
We agree with this critique and have removed reference to refugial populations from this manuscript.

Sincerely,

Kimberly M. Dohms
PhD Candidate, Burg Lab